

# Temporal variation of the relationships between rice yield and climate variables since 1925

Hungyen Chen[1], Yi-Chien Wu[2] and Chih-Yung Teng[2]

[1] Department of Agronomy, National Taiwan University, Taipei, Taiwan
[2] Taichung District Agricultural Research and Extension Station, Council of Agriculture, Changhua, Taiwan

## ABSTRACT

**Background:** Long-term time-series datasets of crop yield and climate variables are necessary to study the temporal variation of climate effects on crops. The aim of this study was to broadly assess assessment of the effects of climate on rice, and the associated temporal variations of the effects during the long-term period.

**Methods:** We conducted field experiments in Taiwan from 1925 to 2019 to collect and analyze rice yield data and evaluate the impacts of changes in average temperature, diurnal temperature range (DTR), rainfall, and sunshine duration on rice yield during cool and warm cropping seasons. We then estimated the relationships between annual grain yield and the climate variables using the time series of their first difference values. We also computed the total relative and annual actual yield changes using regression coefficients for each climate variable for the intervals 1925–1944, 1945–1983, and 1996–2019 to reveal the impacts of climate change on yields and the associated temporal variations during the overall experimental period.

**Results:** The annual daily average temperature calculated from the trend of the regression lines increased by 0.94–1.03 °C during the 95-year period. The maximum temperature remained steady while the minimum temperature increased, leading to decreased DTR. The total annual rainfall decreased by 237–352 mm and the annual total sunshine duration decreased by 93.9–238.9 h during the experimental period. We observed that during the cool cropping season, yield response to temperature change decreased, while that to DTR and rainfall changes increased. During the warm cropping season, all the yield responses to temperature, DTR, and rainfall changes were negative throughout the experimental period. In recent years (1996–2019) the estimated annual actual rice yield changes during the cool cropping season were negatively affected by climate variables (except for sunshine duration), and slightly positively affected (except for temperature) during the warm cropping season. Compared to the effects of temperature and DTR, those of rainfall and sunshine duration on rice yield changes were weak. This study contributes to provide impacts of climate change on rice yield and associated long-term temporal variations over nearly a century.

Corresponding author
Hungyen Chen,
chenhungyen@ntu.edu.tw

## INTRODUCTION

Climate change is a global phenomenon, and its impact on crop yield has been shown to influence the global crop production (*Chmielewski & Potts, 1995*; *Parry et al., 2004*; *Tao et al., 2006*, *2008b*; *Schmidhuber & Tubiello, 2007*; *Lobell et al., 2008*; *Schlenker & Roberts, 2009*; *Zhang & Huang, 2012*; *Kukal & Irmak, 2018*; *Raza et al., 2019*). Research has suggested that for the next 100 years, the global average surface temperature will continue to rise, eventually increasing by 1.1–6.4 °C (*IPCC, 2007*). To assess the benefits of climate change mitigation and agricultural adaptation activities, the response of crop production to climate change has been extensively studied (*Parry, Rosenzweig & Livermore, 2005*; *Chen, 2016*; *Lee & Chen, 2018*; *Ray et al., 2019*). Although many reports highlight the effect of increasing temperature on yield, the impacts of climate change on crop yields vary among studies, even when using the same experimental data (*Peng et al., 2004*; *Sheehy, Mitchell & Ferrer, 2006*). Therefore, comprehensive research analyzing the changes in crop yield in response to interactive climatic factors is imperative (*Lobell & Ortiz-Monasterio, 2007*; *Zhang, Zhu & Wassmann, 2010*; *Steward et al., 2018*). Many studies have reported the impacts of climate change on crop production using climate model projections of average temperature and rainfall (*Adams et al., 1990*; *Rosenzweig & Parry, 1994*; *Parry, Rosenzweig & Livermore, 2005*; *Navarro-Racines et al., 2020*); changes in the diurnal temperature range (DTR, defined as the difference between the daily maximum and minimum temperatures) (*Lobell, 2007*; *Chen, Zhou & Zhou, 2014*), humidity, and solar radiation (*Brown & Rosenberg, 1997*; *Hatfield et al., 2020*); and increased frequency of extreme climatic events (*Rosenzweig et al., 2002*; *White et al., 2006*; *Guntukula, 2020*). To investigate the general response patterns of crop yield change to climate change and variability, a long-term analysis of temporal variations in crop yield and climate variables is crucial (*Lobell, 2007*; *Lobell et al., 2008*). Multiple crop growth simulation models have been used to evaluate the effect of projected climate change on crop yield; however, few studies have investigated the effects of observed climate change on crop yield using consistent long-term field experimental data (*Rosenzweig & Parry, 1994*; *Stooksbury & Michaels, 1994*; *Lobell & Asner, 2003*).

To meet the growing demand for food caused by the worldwide population growth, there is a need to increase global rice production by improving the per unit yields of existing croplands (*Cassman, 1999*; *Tilman et al., 2002*). In Taiwan and many other Asian countries, rice is one of the major grain crops and staple food (*Muthayya et al., 2014*). Taiwan is located at the Tropic of Cancer and has a subtropical climate, characterized by hot and humid summers with a long-day photoperiod and mild to cold winters. The cultivars of japonica rice in Taiwan, which are called Pon-Lai rice, are the only varieties of this rice type that have good-quality and can grow in a subtropical climate at relatively soaring temperatures and short day-length with high yield (*Chang, 1999*; *Lur, Liu & Agrometeorology Section of Central Weather Bureau, 2006*; *Wu, Chang & Lur, 2016*). Owing to the high-quality cultivars and suitable climatic conditions for rice growth in Taiwan, it serves as an ideal location for investigating the rice yield response to climate change in the future (*Shiu, Liu & Chen, 2009*; *Wu, Chang & Lur, 2016*; *Chen et al., 2023*).

In most areas in Taiwan, cultivation occurs over two cropping seasons annually. The cool cropping season starts in February or March and ends in June or July, and the warm cropping season starts in July or early August and ends in November. The patterns of temperature variation during the cool cropping season are the opposite of those during the warm cropping season. The average temperature increases and decreases during the cool and warm cropping seasons, respectively. Thus, it is necessary to analyze the rice yield response to climate variables during the two seasons separately.

Taiwan has experienced a country-wide warming trend since 1900, and both annual and diurnal temperature ranges have also increased (*Hsu & Chen, 2002*). *Lee & Chen (2018)* used regression analysis to classify the annual trends of rice production in various regions of Taiwan from 2003 to 2016 and found that rice production in northern Taiwan showed a downward trend. *Yao & Chen (2009)* used crop growth model software to assess the impact of climate change on rice growth and yield, and the results showed that the average yield of rice cultivated in Taiwan would decrease by 4.7% in 2050. Through a risk analysis model, *Wu et al. (2015)* reported that rice yields in Taiwan are particularly sensitive to temperature, precipitation, and sunlight. *Wang et al. (2023)* utilized a multi-criteria assessment and sensitivity analysis approach, considering factors such as soil, rainfall, temperature, irrigation, and soil erosion, and reported that agricultural lands across Taiwan generally have moderate or high suitability for rice cultivation, especially in southwestern Taiwan. *Promchote et al. (2022)* assessed the effect of winter monsoon variability and climate warming, and their result showed that increased temperatures during the early growth season significantly shortened the rice vegetative phase in all planting dates. *Chen et al. (2023)* analyzed the long-term yield of rice in Taiwan as well as the temporal trends of reference evapotranspiration and crop water status, showing that the impact of water-deficit stress has increasingly affected rice growth in recent years.

The long-term of the effects of climate on crops are garnering increasing attention from agronomists. However, current studies on temporal variations of the climatic effects on rice lack consistent long-term time series. Our goal was to provide a broad assessment of the effects of climate on the major global crop, rice (Oryza spp.), throughout the major cropping seasons, and the associated temporal variations of the effects during a 95-year period. In the present study, we analyzed the yield data of 14 rice cultivars from field experiments conducted under irrigated conditions and optimal management at a research farm in Taichung, Taiwan, from 1925 to 2019 to evaluate the effects of changes in climate variables, including average temperature, DTR, rainfall, and sunshine duration on rice yield in both cool and warm cropping seasons.

## MATERIALS AND METHODS

First, weather data was collected at the farm to deduce long-term temporal trends of average temperature, rainfall, and sunshine duration during the 95-year period. Second, a multiple linear regression model was applied to evaluate the relationships between annual grain yield and climate variables using the time series of their first difference values. Third, the total relative and annual actual yield changes, computed separately from percent and actual regression coefficients for each climate variable for the intervals 1925–1944,

1945–1983, and 1996–2019 were used to reveal the impacts of climate change on rice yields and their temporal variations during the experimental period.

## Field experiment

A long-term field experiment on rice growth was conducted between 1925 and 2019 at the research farm in the Taichung District Agricultural Research and Extension Station, Council of Agriculture, Executive Yuan, Taiwan (1925–1983: 24°09′N 120°41′E, altitude 77 m above mean sea level; 1996–2019: 24°00′N 120°32′E, altitude 19 m above mean sea level). Data were collected as previously described in *Chen et al. (2023)*. In 1984, the research station and farm were moved to a location approximately 20 km from the original location. The field site was located on the coastal plain of western Taiwan, where the soil is covered with alluvial material from the central mountain range, which is the principal mountain range on the Taiwan island. The parent materials of the soil were limestone, mudstone, and clay slate. The surface soil was light yellow in color and fertile for cultivation with a pH of 7.43.

The long-term research was conducted across two cropping seasons (cool and warm) for the experimental time period and measured the crop yields by multiple researchers at the research station. The data of the crop yields from 1996 to 2019 were measured by the authors and researchers from the authors' research team and the data from 1925 to 1983 were obtained from the records of the database at the research station. On the one hand, seeds of the cool cropping season rice were sown in mid-January, and seedlings were dibbled either in late February or early March every year from 1925 to 2019 except in 1948–1951, 1985–1995, and 2014–2016. The cool cropping season rice was harvested in either late June or early July. On the other hand, seeds of the warm cropping season rice were sown at the end of June, and seedlings were dibbled either in late July or early August every year from 1925 to 2019, except for 1945, 1947–1951, 1985–1995, and 2013–2015. The rice of this season was harvested in either late October or early November. Rice seeds were initially grown in a nursery box, following which, sprouted seedlings were transplanted into the field by hand dibbling. The area of plots for each cultivar was 27 m$^2$ (3 m × 3 m × 3 plots). Four to six seedlings were dibbled in each hole; the rows were 30 cm wide and arranged in 15 cm intervals. The grain yield was obtained by harvesting all the crops in the hills in the plots (at a grain maturity rate of 98%), except for those in the side rows, and by measuring their grain weight.

A base fertilizer, with a nitrogen:phosphorus:potassium ratio (N:P:K) of 12:7.87:9.96, was added to the soil at a rate of 200 kg ha$^{-1}$. The top dressing of fertilizer application was performed three times at 10–15, 20–30, and 55–65 days and at 7–10, 14–21, and 45–55 days after transplanting in the cool and warm cropping seasons, respectively. The top three fertilizers were added to the soil at a rate of 200 (N:P:K = 21:0:0), 200 (N:P:K = 12:5.68:10.79), and 150 kg ha$^{-1}$ (N:P:K = 12:5.68:10.79) during both cropping seasons. Herbicides were applied during the cropping seasons, and insecticides were applied after checking the rice for symptoms of pests and diseases in rice.

Fourteen rice cultivars were used throughout the experimental period during the cool and warm cropping seasons, individually. In cool cropping season, Nakamura

(NM; 1925–1931), Taichung S2 (TCS2; 1925–1932), Baiker (BK; 1925–1944), Taichung S6 (TCS6; 1933–1944), Wugen (WG; 1925–1947, 1952–1976), Baimifun (BMF; 1945–1947, 1952–1976), Taichung 65 (TC65; 1930–1947, 1952–1983), Taichung 150 (TC150; 1945–1947, 1952–1983), Taichung Indica 1 (TCI1; 1964–1983), Taichung Indica 3 (TCI3; 1977–1983), Taiagro 67 (TA67; 1996–2013, 2017–2019), Taichung 189 (TC189; 1996–2013, 2017–2019), Taichung Indica 10 (TCI10; 1996–2013, 2017–2019), and Tai Japonica 9 (TJ9; 2000–2013, 2017–2019) were used. In warm cropping season, Nakamura (NM; 1925–1931), Taichung S2 (TCS2; 1925–1944), Jingou (JG; 1925–1944), Nyaoyao (NY; 1925–1944), Swanjian (SJ; 1946, 1952–1976), Sianlou (SL; 1946, 1952–1976), Taichung 65 (TC65; 1930–1944, 1946, 1952–1983), Taichung 150 (TC150; 1946, 1952–1983), Taichung Indica 2 (TCI2; 1977–1983), Taichung Indica 3 (TCI3; 1977–1983), Taiagro 67 (TA67; 1996–2012, 2016–2019), Taichung 189 (TC189; 1996–2012, 2016–2019), Taichung Indica 10 (TCI10; 1996–2012, 2016–2019), and Tai Japonica 9 (TJ9, 2000–2012, 2016–2019) were used. Data were collected as previously described in *Chen et al. (2023)*. During the experimental period, the data of the crop yield were not available in 1948–1951, 1985–1995, 2014, and 2015 for any cultivar, because the field experiments were not conducted in those years.

## Weather data

A weather station was set up at the farm in the research site, which was surrounded by field crops and had a flat topography. Recording daily weather data began on January 1, 1925. Meteorological instruments at the station include a solarimeter, psychrometer, thermohygrograph, and glass thermometers for minimum and maximum temperatures. Air temperature, rainfall, and sunshine duration were measured during the cropping seasons throughout the experimental period and used for analyses. Data were collected as previously described in *Chen et al. (2023)*. The data of the weather variables collected at the weather station were calibrated any quality controlled by replacing the missing value and outlier by the mean of the values in the preceding and following years. In total, the percentage of the missing daily weather data is 2.65% (882 days/33,237 days) during the experimental period. Mean imputation is very simple to understand and to apply, and do not reduce the sample size. However, mean substitution leads to bias in standard errors, variance, and multivariate estimates such as correlation or regression coefficients (*Kalton & Kasprzyk, 1986*). For air temperature, the daily average, minimum, and maximum temperatures throughout the cultivation period were measured from January 20 to June 23 during the cool cropping season and from July 4 to November 22 during the warm cropping season. The daily average temperature (T) for each year was calculated as the average of the daily temperature values during the two cropping seasons. The DTR for each year was calculated as the average difference between the daily maximum and minimum temperatures in the two cropping seasons. The total rainfall (R) and sunshine duration (S), during the cultivation period were calculated as the sum of their respective daily values from January 20 to June 23 during the cool cropping season and from July 4 to November 22 during the warm cropping season.

### Yield and climate variable models.

A simple linear regression equation was used to determine the linear time trend of each climate variable. The underlying relationship between $x$ and $t$ can be described by:

$$x = \beta_0 + \beta_1 t + \varepsilon \tag{1}$$

where x is the climate variable, t is the year corresponding to x, $\beta_0$ is the intercept, $\beta_1$ is the regression coefficient representing the rate of change of climate variable per year, and $\varepsilon$ is the model error.

To eliminate the influence of technological trends on rice grain yield, a time series of the first difference was computed for the yield and climate variables by subtracting their previous year's value from those of the subsequent year (*Nicholls, 1997*; *Lobell, 2007*; *Chen, Zhou & Zhou, 2014*). A multiple linear regression model was used to describe the relationship between crop yield and climate variables by considering the first difference value of grain yield ($\Delta Y$) as the response variable and those of climate variables ($\Delta T$, $\Delta DTR$, $\Delta R$, and $\Delta S$) as explanatory variables for each rice cultivar in each cropping season. This is represented by the following:

$$\Delta Y = \beta_0 + \beta_T \Delta T + \beta_{DTR} \Delta DTR + \beta_R \Delta R + \beta_S \Delta S + \varepsilon \tag{2}$$

where $\beta_0$ is the model intercept, $\beta$ is the regression coefficient for each climate variable, and $\varepsilon$ is the model error.

The percentage regression coefficient for each climate variable was computed as the $\beta$ divided by the average grain yield during the cultivation period for each rice cultivar as follows:

$$\beta_\% = \frac{\beta}{\text{Average grain yield}} \tag{3}$$

where $\beta_\%$ and $\beta$ are the percentage and absolute regression coefficients of the climate variables ($\Delta T$, $\Delta DTR$, $\Delta R$, and $\Delta S$), respectively.

The estimated total relative and annual actual yield changes were calculated using $\beta_\%$ and $\beta$ during the cultivation period.

Total relative yield change (%) was calculated as:

$$\text{Total relative yield change} = \beta_\% \times \Delta \text{Climate variable}_{\text{Total years}} \tag{4}$$

Annual actual yield change (kg ha$^{-1}$) was calculated as:

$$\text{Annual actual yield change} = \frac{\beta \times \Delta \text{Climate variable}_{\text{Total years}}}{\text{Number of years}} \tag{5}$$

We considered temperature, rainfall, and sunshine duration for our analyses as previous reports have suggested examining all climatic factors when evaluating the effects of climate change on rice yield (*Lobell & Ortiz-Monasterio, 2007*; *Zhang, Zhu & Wassmann, 2010*; *Guntukula, 2020*). A positive percentage regression coefficient reflects a positive correlation between grain yield and climate variables, and a negative coefficient indicates a negative relationship between the two factors.

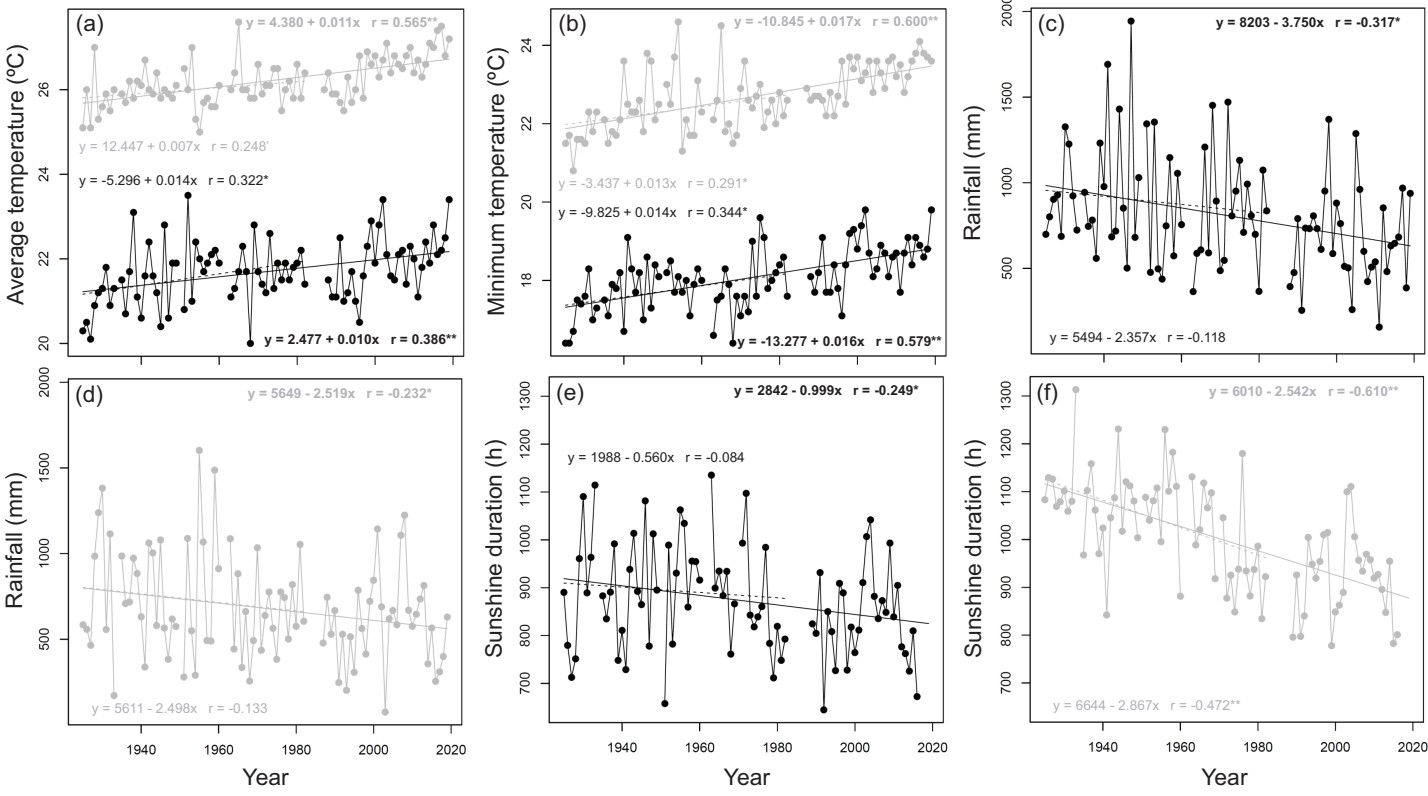

**Figure 1 Annual trend of air temperature, rainfall, and sunshine duration during the experimental period.** Mean daily (A) average and (B) minimum temperatures. (C and D) Total rainfall. (E and F) Total sunshine duration. Black and grey colors represent the cool and warm cropping seasons, respectively. Lines in the panels represent linear regression line. Solid lines and bold-typed equations represent the 1925–2019 time period. Dash lines and non-bold-typed equations represent the 1925–1983 time period. **, *, and ' represent $p$-value $< 0.01$, $0.01 < p$-value $< 0.05$, and $0.05 < p$-value $< 0.1$, respectively.

## RESULTS

### Climate change during the rice cropping seasons

From 1925 to 2019, the annual daily average ranged from 20 (in 1968) to 23.5 °C (in 1952), with an average value of 21.7 ± 0.7 °C (s.d.) during the cool cropping season and ranged from 25 (in 1955) to 27.6 °C (in 1965), with an average value of 26.2 ± 0.6 °C (s.d.) during the warm cropping season (Fig. 1A). DTR values ranged from 7.0 (in 1951 and 1990) to 11.7 °C (in 1968), with an average value of 8.6 ± 0.9 °C (s.d.) during the cool cropping season and ranged from 4.6 (in 1965) to 11.4 °C (in 1954), with an average value of 8.6 ± 1.0 °C (s.d.) during the warm cropping season. The total annual rainfall ranged from 157 (in 2011) to 1,945 mm (in 1947), with an average value of 807 ± 335 mm (s.d.) during the cool cropping season (Fig. 1C) and ranged from 75 (in 2003) to 1,602 mm (in 1955), with an average value of 680 ± 307 mm (s.d.) during the warm cropping season (Fig. 1D). The annual total sunshine duration ranged from 644.7 (in 1992) to 1,393.0 h (in 2018), with an average value of 884.1 ± 131.6 h (s.d.) during the cool cropping season (Fig. 1E) and ranged from 720.5 (in 2016) to 1,319.2 h (in 2019), with an average value of 1,006.4 ± 124.0 h (s.d.) during the warm cropping season (Fig. 1F).

Throughout the experimental period, the annual daily average and minimum temperatures calculated from the trend of the regression lines increased by 0.94 and 1.50 °C, respectively, during the cool cropping season and by 1.03 and 1.60 °C, respectively, during the warm cropping season (Figs. 1A and 1B). The annual daily maximum temperature calculated from the trend of the regression lines increased by 0.28 °C during the cool cropping season and decreased by 0.09 °C during the warm cropping season from 1925 to 2019. During the experimental period, changes in the maximum temperature were limited and the linear time trends were not significant. Throughout the experimental period, the total annual rainfall calculated from the trend of the regression lines in the cool and warm cropping seasons decreased by 352 and 237 mm, respectively (Figs. 1C and 1D). During the experimental period, the decrease in rainfall during the cool cropping season was 1.5 times greater than that in the warm cropping season. Throughout the experimental period, the annual total sunshine duration calculated from the trend of the regression lines during cool and warm cropping seasons decreased by 93.9 and 238.9 h, respectively (Figs. 1E and 1F). During the experimental period, the decrease in sunshine duration during the warm cropping season was 2.5 times greater than that in the cool cropping season. The slope (regression coefficient, $\beta_1$) of the simple linear regression lines are significant ($p$-value < 0.05) for all the climate variables during 1925–2019 (Fig. 1).

**Impacts of climate change on rice yield**

The annual rice yield ranged from 9,500 (TCI3 in 1978) to 1,530 kg ha$^{-1}$ (TCS6 in 1935), with an average value of 5,532 ± 1,337 kg ha$^{-1}$ (s.d.) during the cool cropping season and ranged from 9,363 (TCI10 in 1984) to 2,180 kg ha$^{-1}$ (SJ in 1969), with an average value of 4,523 ± 1,072 kg ha$^{-1}$ (s.d.) during the warm cropping season.

The long-term temporal variations in the estimated impacts of the climate variables on the grain yield of the 14 rice cultivars during the cool and warm cropping seasons from 1925 to 2019 are shown in Fig. 2. For climate variables, their first difference values were weakly correlated (0.3 < | r | ≤ 0.5) with each other during the cool cropping season and showed little correlation (| r | ≤ 0.3) during the warm cropping season (Table 1). Moreover, different rice cultivars were used throughout the study duration (Fig. 2). For each cropping season, four groups of cultivars with overlapping cultivation periods were clustered to calculate the average group value based on their cultivation period, which represents the impact of climate variables on the rice yield during each time period. Based on the cultivation period, four, four, two, and four cultivars were grouped and analyzed during the 1925–1944, 1945–1983, 1977–1983, and 1996–2019 time periods, respectively (Fig. 2). Group usage for these time periods was previously described in *Chen et al. (2023)*. The annual variations in the yields of the cultivars with overlapping cultivation periods were correlated with each other in the same groups (Fig. 3). As the 1977–1983 period included only two cultivars, spanned only seven years, and overlapped with the 1945–1983 period, it was excluded and only the remaining three long-term time periods were included in subsequent analyses.

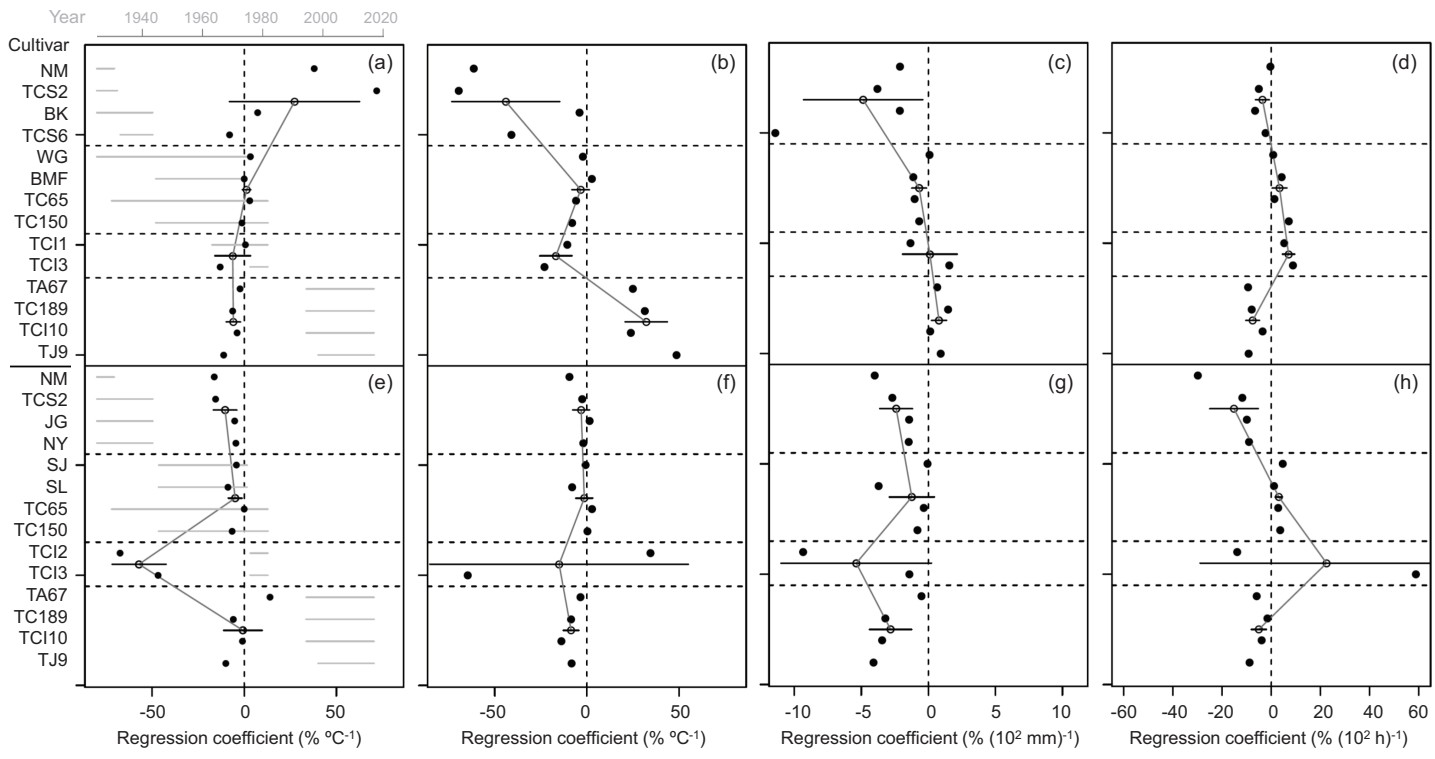

**Figure 2 Percent regression coefficient of rice yield response to climate variables.** Filled and open circles represent the value of a cultivar and the average value of a group of cultivars having overlapping cultivation periods, respectively. The length of black lines on open circles represent two standard deviations. Horizontal dashed lines separate different groups of cultivars having overlapping cultivation periods. The length of grey lines represents the cultivation period of a cultivar. (A and E) Average temperature. (B and F) Diurnal temperature range (DTR). (C and G) Rainfall. (D and H) Sunshine duration. (A–D) Cool cropping season. (E–H) Warm cropping season. The values of percent regression coefficient of rice yield response to climate variables are showed in Table 2.

**Table 1 Correlation coefficients between pairs of climate variables during cool and warm cropping seasons.**

| | | Cool cropping season | | | | | Warm cropping season | | |
|---|---|---|---|---|---|---|---|---|---|
| | Climate variable | $\Delta T$ | $\Delta DTR$ | $\Delta R$ | | Climate variable | $\Delta T$ | $\Delta DTR$ | $\Delta R$ |
| Cool cropping season | $\Delta DTR$ | 0.282** | – | – | Warm cropping season | $\Delta DTR$ | −0.220* | – | – |
| | $\Delta R$ | −0.403** | −0.308** | – | | $\Delta R$ | −0.179 | −0.274* | – |
| | $\Delta S$ | 0.426** | 0.496** | −0.561** | | $\Delta S$ | −0.038 | 0.105 | −0.186 |

**Note:**
$\Delta T$, $\Delta DTR$, $\Delta R$, and $\Delta S$ represent the first difference values of average temperature, diurnal temperature range, rainfall, and sunshine duration, respectively. ** and * Represent $p$-value < 0.01 and 0.05 < $p$-value < 0.1, respectively.

Determining the average percentage regression coefficients of $\Delta T$ in the cool cropping season revealed positive average values in 1925−1944 and 1945−1983, but a negative average value in 1996−2019 (Fig. 2A and Table 2); moreover, these values decreased throughout the experimental period. In the warm cropping season, these coefficients were negative in all time periods (Fig. 2E and Table 2). In the cool cropping season, the average percent regression coefficients of $\Delta DTR$ increased throughout the experimental period and had negative average values in 1925−1944 and 1945−1983, but a positive average value in 1996−2019 (Fig. 2B and Table 2). In the warm cropping season, the average percentage

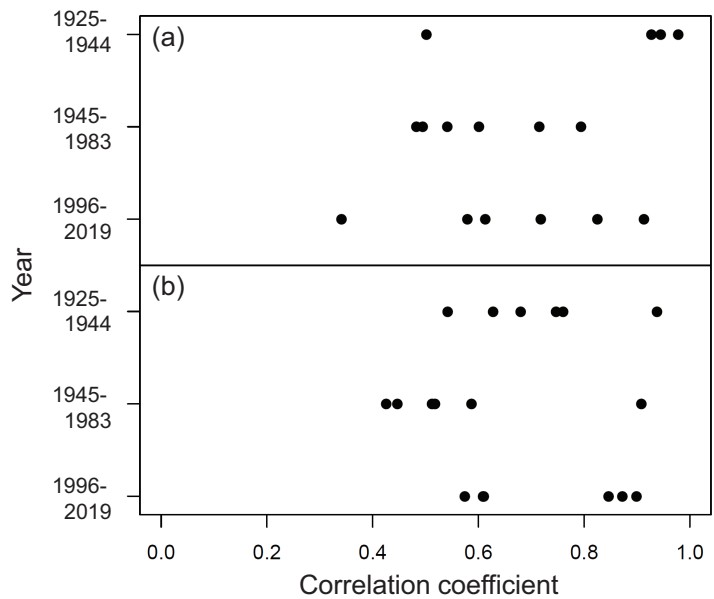

**Figure 3 Correlation coefficients between the yields of cultivar pairs in the same groups with overlapping cultivation periods.** (A) Cool and (B) warm cropping seasons.

regression coefficients of DTR were positive in all time periods (Fig. 2F and Table 2). In the cool cropping season, the average percent regression coefficients of ΔR increased throughout the experimental period and had negative average values in 1925–1944 and 1945–1983, but a positive average value in 1996–2019 (Fig. 2C and Table 2). In the warm cropping season, the average percent regression coefficients of ΔR were positive for all time periods (Fig. 2G and Table 2). Additionally, the average percent regression coefficients of ΔS in both cool and warm cropping seasons revealed negative average values in 1925–1944 and 1996–2019, but a positive average value in 1945–1983 (Figs. 2D, 2H and Table 2). The average percent regression coefficients of ΔS in both cropping seasons increased within the period of 1925–1983, but decreased after 1996 (Figs. 2D, 2H and Table 2).

## Actual yield changes in response to climate variables

The mean total relative yield change in the cultivars related to climate variables in cool and warm cropping seasons during 1925–1944, 1945–1983, and 1996–2019 is showed in Table 3. The relative yield change could be transformed into actual yield change by incorporating the average grain yield of the rice cultivar (Fig. 4).

In the cool cropping season, the annual actual yield change in the cultivars associated with temperature in cool cropping seasons during 1925–1944, 1945–1983, and 1996–2019 were 11.8, 0.6, and −3.7 kg ha$^{-1}$, respectively (Fig. 4A), whereas that in the warm cropping seasons during the three periods were −4.2, −2.1, and −0.4 kg ha$^{-1}$, respectively (Fig. 4A). For DTR, the annual yield change in the three periods during the cool cropping season were found to be 23.3, 2.1, and −25.9 kg ha$^{-1}$, respectively, and 1.9, 0.9, and 7.0 kg ha$^{-1}$, respectively, during the warm cropping season (Fig. 4B). For the annual yield associated

**Table 2 Percent regression coefficient (%) of rice yield response to climate variables.**

| Cool cropping season | | | | | Warm cropping season | | | | |
|---|---|---|---|---|---|---|---|---|---|
| Cultivar | ΔT | ΔDTR | ΔR | ΔS | Cultivar | ΔT | ΔDTR | ΔR | ΔS |
| NM | 37.94 | −61.38 | −2.12 | −0.35 | NM | −16.28 | −9.45 | −4.00 | −29.87 |
| TCS2 | 71.82 | −69.51 | −3.80 | −5.04 | TCS2 | −15.55 | −2.51 | −2.69 | −11.80 |
| BK | 7.28 | −3.99 | −2.13 | −6.61 | JG | −5.22 | 1.50 | −1.44 | −9.95 |
| TCS6 | −8.00 | −40.92 | −11.42 | −2.36 | WY | −4.57 | −1.95 | −1.47 | −9.07 |
| Mean | 27.26 | −43.95 | −4.87 | −3.59 | Mean | −10.41 | −3.10 | −2.40 | −15.17 |
| WG | 3.29 | −2.14 | 0.09 | 0.80 | SJ | −4.30 | −0.73 | −0.06 | 4.66 |
| BMF | 0.00 | 2.68 | −1.13 | 4.21 | SL | −8.85 | −7.99 | −3.71 | 1.13 |
| TC65 | 2.98 | −5.90 | −1.03 | 1.25 | TC65 | −0.03 | 2.77 | −0.35 | 2.80 |
| TC150 | −1.30 | −8.01 | −0.68 | 7.13 | TC150 | −6.57 | 0.27 | −0.81 | 3.64 |
| Mean | 1.24 | −3.34 | −0.69 | 3.35 | Mean | −4.94 | −1.42 | −1.23 | 3.06 |
| TCI1 | 0.61 | −10.64 | −1.33 | 5.27 | TCI2 | −67.40 | 34.51 | −9.34 | −13.88 |
| TCI3 | −13.08 | −22.99 | 1.56 | 8.82 | TCI3 | −46.75 | −64.65 | −1.41 | 58.85 |
| Mean | −6.23 | −16.82 | 0.11 | 7.04 | Mean | −57.08 | −15.07 | −5.38 | 22.49 |
| TA67 | −2.32 | 24.94 | 0.65 | −9.46 | TA67 | 13.96 | −3.49 | −0.52 | −5.96 |
| TC189 | −6.29 | 31.44 | 1.47 | −7.99 | TC189 | −5.93 | −8.56 | −3.21 | −1.57 |
| TCI10 | −3.88 | 23.83 | 0.14 | −3.55 | TCI10 | −0.98 | −13.86 | −3.45 | −3.91 |
| TJ9 | −11.19 | 48.61 | 0.92 | −9.24 | TJ9 | −10.02 | −8.33 | −4.09 | −8.83 |
| Mean | −5.92 | 32.20 | 0.79 | −7.56 | Mean | −0.74 | −8.56 | −2.82 | −5.07 |

Note:

ΔT, ΔDTR, ΔR, and ΔS represent the first difference values of average temperature, diurnal temperature range, rainfall, and sunshine duration, respectively.

**Table 3 Total relative yield change (%) in response to climate variables during the cool and warm cropping seasons in the 1925–1944, 1945–1983, and 1996–2019 time periods.**

| | 1925-1944 | | | | 1945–1983 | | | | 1996–2019 | | | |
|---|---|---|---|---|---|---|---|---|---|---|---|---|
| | Cool cropping season | | Warm cropping season | | Cool cropping season | | Warm cropping season | | Cool cropping season | | Warm cropping season | |
| Climate variable | Mean | SD | Mean | SD | Mean | SD | Mean | SD | Mean | SD | Mean | SD |
| ΔT | 5.2 | 6.8 | −2.2 | 1.3 | 0.5 | 0.9 | −2.1 | 1.6 | −1.4 | 0.9 | −0.2 | 2.7 |
| ΔDTR | 10.8 | 7.2 | 1.1 | 1.6 | 1.6 | 2.3 | 1.0 | 3.1 | −9.6 | 3.4 | 3.5 | 1.7 |
| ΔR | 3.5 | 3.2 | 1.1 | 0.6 | 1.0 | 0.8 | 1.2 | 1.6 | −0.7 | 0.5 | 1.6 | 0.9 |
| ΔS | 0.2 | 0.2 | 5.7 | 3.7 | −0.4 | 0.3 | −2.3 | 1.1 | 0.5 | 0.2 | 2.3 | 1.4 |

Note:

ΔT, ΔDTR, ΔR, and ΔS represent the first difference values of average temperature, diurnal temperature range, rainfall, and sunshine duration, respectively. Mean and SD represent the average value and standard deviation of a group of cultivars having overlapping cultivation periods.

with rainfall during the cool cropping season in the three periods are 7.1, 1.2, and −1.8 kg ha$^{-1}$, respectively, and 2.2, 1.2, and 3.2 kg ha$^{-1}$, respectively, for the warm cropping season (Fig. 4C). For sunshine duration, the annual yield change during the cool cropping season in the three periods are 0.5, −0.5, and 1.4 kg ha$^{-1}$, respectively, and 10.7, −2.4, and 4.6 kg ha$^{-1}$, respectively, during the warm cropping season (Fig. 4D).

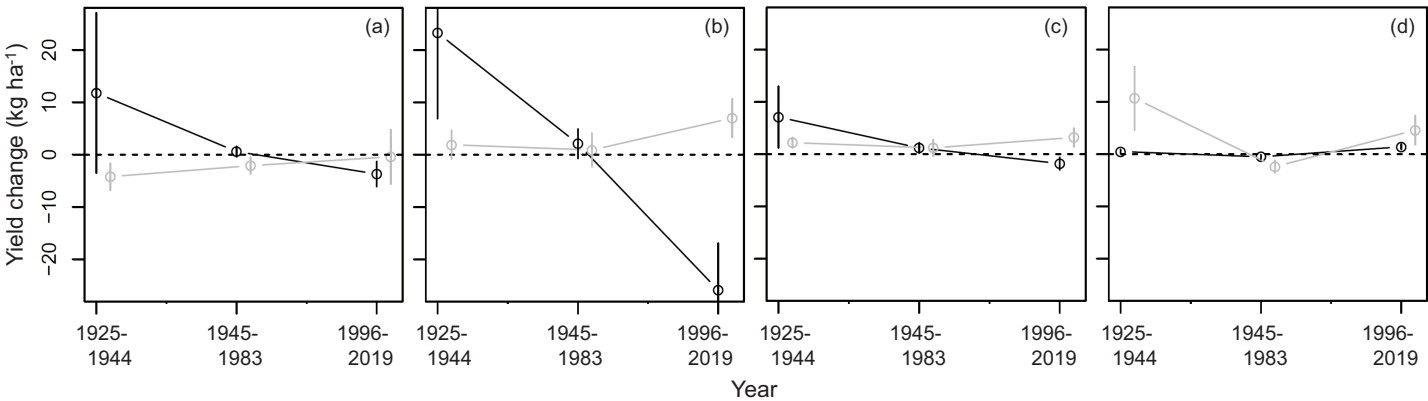

**Figure 4 Annual actual yield change associated with climate variables during 1925–1944, 1945–1983, and 1996–2019 in cool and warm cropping seasons.** (A) Average temperature; (B) DTR; (C) rainfall; (D) sunshine duration. Open circles represent the average value of a group of cultivars having overlapping cultivation period. The length of vertical lines on open circles represent the value of two standard deviations. Black and grey colors represent the cool and warm cropping seasons, respectively.

## DISCUSSION

The global annual mean surface air temperature rose by approximately 1.05–1.35 °C between 1925 and 2019 (*Lenssen et al., 2019*; *GISTEMP Team, 2022*), which is consistent with our observations. The increase in average temperature during the warm cropping season was 0.09 °C greater than that during the cool cropping season for the 95-year period. A rise in temperature, especially during summer, has been reported in some areas, which could be due to the soil moisture-dependent land-temperature and land-precipitation positive feedback processes (*Bartolini et al., 2012*; *Lorenz, Jaeger & Seneviratne, 2010*). The minor change in maximum temperature observed in our study is consistent with the results of a previous study conducted during the dry and wet seasons at the research farm of the International Rice Research Institute in the Philippines (*Peng et al., 2004*). It has been reported that steady day temperatures are accompanied with increasing night temperatures, leading to decreased DTR. During the cropping season, a decrease in DTR leads to warmer nights, which is harmful for plant growth as it causes an increase in respiration rate (*Leopold & Kriedemann, 1975*). Substantial decreasing trends in DTR have been observed globally (*Easterling et al., 1997*; *Vose, Easterling & Gleason, 2005*), and certain climate models have been used to project further significant changes in DTR (*Stone & Weaver, 2003*; *Lobell, 2007*). A decreasing trend in annual rainfall has been observed in some areas globally (*Kumar, Jain & Singh, 2010*; *Nisansala et al., 2019*). In Taiwan, there is increasing rainfall decline in the cool season than in the warm season, which could be associated with the decrease in the number of rainy days and rainfall during the rainy season. Our results also suggested that the dry and wet seasons could be more distinct in the future. The decreasing trends in sunshine duration observed in multiple countries (*Kaiser & Qian, 2002*; *Stanhill & Cohen, 2005*; *Jhajharia & Singh, 2011*) could be an indication of increased haze pollution (*Kaiser & Qian, 2002*), especially during summer.

The research farm in the present study was moved approximately 20 km from its original location in 1984. Trends of increase in temperature and decrease in rainfall and

sunshine duration observed from 1925 to 1983 are similar to those observed from 1925 to 2019 (Fig. 1). The differences in slopes for temperature, rainfall, and sunshine duration in the cool cropping season between 1925–2019 and 1925–1984 were −0.004 °C, −1.393 mm, and −0.439 h per year, respectively, whereas those observed during the same time frame in the warm cropping season were 0.004 °C, −0.021 mm, and 0.325 h per year, respectively (Fig. 1). The minor difference detected between the two locations may be due to the different altitudes and rate of climate change during the 95-year period (*Lenssen et al., 2019*; *GISTEMP Team, 2022*). Moreover, the rise in global average temperature over the last 50 years has been nearly twice of that over the last 100 years (*IPCC, 2007*).

Our result is consistent with studies based on process-based models that report a negative response of crop yields to global warming in the absence of other climatic variables (*Rosenzweig & Parry, 1994*; *Leng & Hall, 2020*). As the changes in maximum and minimum temperatures tend to be highly correlated from year to year, the range of interannual variations in DTR could be small (*Lobell, 2007*). This may lead to uncertainty in ΔDTR estimation in an empirical model. However, the observations at our research station showed a clear increasing trend for minimum temperature, but only a minor change in the maximum temperature over long-term.

The correlations among the changes in climate variables may make it difficult to distinguish the effects of individual climate variables in an empirical model owing to co-linearity (*Lobell, 2007*). Changes in rainfall have often been correlated with those in sunshine duration and DTR since the cloud cover during rainy days may reduce values of the other two variables (*Dai, Trenberth & Karl, 1999*). Moreover, changes in sunshine duration are often correlated with those in temperature. Higher sunshine duration is associated with increased DTR and could result in longer growth duration of crops (*Lobell & Ortiz-Monasterio, 2007*). Contrary to these correlations reported in literature, the observations in our study showed low to little correlation among the climate variables during the cropping seasons (Table 1).

From 1925 to 1944, we determined the positive yield responses for all climate variables, except for temperature in the warm cropping season. A positive yield response to temperature in the cool cropping season may suggest that the average temperature is within the optimal temperature range for rice production and that a warming trend could increase rice yield (*Yang et al., 2008*; *Liu et al., 2010*). High temperature (37°/27 °C) after main crop cutting resulted in high spikelet sterility in the ratoon crop in the International Rice Research Institute in Philippines (*Chauhan, Lopez & Vergara, 1990*). In our study, temperature contributed negatively to rice yield during the warm cropping season throughout the experimental period. A negative yield response to temperature in the warm cropping season suggests that the temperature might be above the optimal temperature for rice production and that a warming trend could reduce rice yield (*Peng et al., 2004*; *Tao et al., 2006*; *Sheehy, Mitchell & Ferrer, 2006*; *Chen, 2016*). During the cool season, yield changes, which were caused by the increases in temperature, DTR, and rainfall, varied from increase to decrease from 1925 to 2019. The range of yield change in response to sunshine was wider in the warm cropping season than that in the cool cropping season. A study conducted using experimental data from the Philippines revealed that a combined

effect of decreasing solar radiation and increasing minimum temperature decreased the rice yield (*Sheehy, Mitchell & Ferrer, 2006*). During the warm season, a reverse pattern in yield change responses to temperature was observed during the 1925 to 2019 period. We also inferred positive effects of DTR and rainfall on rice yield during the warm cropping season. Some studies have revealed that rice yield may be affected by temperature and precipitation, resulting from physiological mechanisms (*Tao et al., 2006*, *2008a*).

From 1996 to 2019, during the cool cropping season, we observed negative effects of climate variables (except for sunshine duration) on annual rice yield change, while during the warm cropping season, minor positive effects (except for temperature) were found. The negative effects of increased DTR have been reported in some studies on rice yields (*Tao et al., 2006*; *Lobell, 2007*), and spikelet sterility for rice has been positively correlated with average maximum temperature (*Tao et al., 2006*). In addition, an increase in DTR may reduce yield because increasing the maximum temperature leads to increased water stress and reduced net photosynthetic rates (*Dhakhwa & Campbell, 1998*; *Tao et al., 2006*). The negative effects of increased rainfall may be due to the humid climate and excessive precipitation observed during the rice heading period, which could lead to yield loss from diseases, insects, and insufficient solar radiation (*Tao & Yokozawa, 2005*).

In our study, the data were collected from the same research farm by the same research station since 1925. During more than 90 years of experimentation, it is impossible to maintain the same environmental and cultivation conditions. Although the farm was relocated in 1984, it was 20 km away from the original location; therefore, the same pattern of climate change was recorded between the two sites. Simple linear regression lines for 1925–2019 and 1925–1983 have consistent slopes (Fig. 1). To obtain the crop yield response to global or national climate change, large-scale research should be conducted. One of the uncertainties in our results is that data collected on different spatial and temporal scales or from different areas may produce varied outcomes (*Tao et al., 2006*; *Liu et al., 2012*). In this study, long-term temporal variation in the rice yield response to climate variables was revealed, even though the rice cultivars changed throughout the experimental period. In such long-term studies, it is almost impossible to maintain consistency in using the same cultivars for crop yield experiments and even harder for the national crop production data collected from farmers. A total of 14 rice cultivars were used during the trial period. The planting periods of each cultivar were different, and it can be difficult to compare the long-term climatic effects of each cultivar. In addition, cultivars used in the early period of the experiments are no longer cultivated today. It is also difficult to thoroughly account for other factors that may affect crop growth, such as soil fertility, insects, disease, and plant density (*Altieri & Nicholls, 2003*; *Chen, Yamagishi & Kishino, 2014*; *Chen et al., 2019*; *Chen, 2019*; *Li, Dai & Chen, 2022a*, *2022b*); as well as for human-induced effects, such as improving technology, modern management, and differences in practices of cultivators (*Chen, 2018*, *2019*), especially for over nine decades of observations. Nowadays, climate change also includes the increasing frequency of extreme climatic events, which results in frequent agricultural meteorological disasters. Future studies should consider the effects of such events on rice yield. Moreover, to improve insights on the important issue of climate's impact on crop production, other quantitative

methods have also been suggested, including an agricultural production cycle model combined with historical data and a decomposition simulation approach (*Dixon & Rimme, 2002*).

## CONCLUSIONS

This study revealed the long-term impact of climate change on rice yield by analyzing the effect of changes in average temperature, DTR, rainfall, and sunshine duration during cool and warm rice cropping seasons. The average temperature calculated from the trend of the regression lines increased by 0.94–1.03 °C during the 95-year period. The maximum temperature remained steady while the minimum temperature increased, leading to decreased DTR. Moreover, the results showed decreasing trends for rainfall and sunshine duration climate variables during the experimental period. Estimating first difference values revealed that in the cool cropping season, the yield response to $\Delta T$ decreased, whereas that to $\Delta DTR$ and $\Delta R$ increased; in the warm cropping season, yield responses to $\Delta T$, $\Delta DTR$, and $\Delta R$ were negative throughout the experimental period. For the estimated actual yield changes in recent years (1996–2019), negative effects of climate variables (except for sunshine duration) on changes in annual rice yield were found in the cool cropping season, whereas slight positive effects (except for temperature) were observed in the warm cropping season. Compared to the effects of temperature and DTR on rice yield, those of rainfall and sunshine duration were weaker. In the future, the impact of climate change on rice yield reported in this study must be considered in addition to other adaptation strategies targeting breeding technologies and agronomic efforts to maintain high-quality rice productivity.

## ACKNOWLEDGEMENTS

The authors wish to thank Dr. Jia-Ling Yang, Ms. Chia-Chi Cheng, and other researchers in Taichung District Agricultural Research and Extension Station, Council of Agriculture, Taiwan who assisted in the field investigation and data collection.

### Funding

This work was supported by funding from the National Science and Technology Council, Taiwan (109-2313-B-002-027-MY3) to Hungyen Chen. The funders had no role in study design, data collection and analysis, decision to publish, or preparation of the manuscript.

### Grant Disclosures

The following grant information was disclosed by the authors:
National Science and Technology Council, Taiwan: 109-2313-B-002-027-MY3.

### Competing Interests

The authors declare that they have no competing interests.

## Author Contributions

- Hungyen Chen conceived and designed the experiments, performed the experiments, analyzed the data, prepared figures and/or tables, authored or reviewed drafts of the article, and approved the final draft.
- Yi-Chien Wu performed the experiments, prepared figures and/or tables, and approved the final draft.
- Chih-Yung Teng performed the experiments, prepared figures and/or tables, and approved the final draft.

## Data Availability

The data that supports the findings of this study are available in the Supplemental Files.

## Supplemental Information

Supplemental information for this article can be found online at http://dx.doi.org/10.7717/peerj.16045#supplemental-information.

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
