# Peer review of "Temporal variation of the relationships between rice yield and climate variables since 1925"

_PeerJ, doi:10.7717/peerj.16045_

## Round 0.1 · original submission · Major Revisions

Major revisions are required and improve paper quality through new citations and new references, improve English, improve abstract, and discussion. Conclusions must be based on the results obtained, thanks.

Reviewer 1 ·

Basic reporting

The paper is well-organized and well-written. Raw data shared. Some suggestions about figures and other sections of the paper are provided below.

Experimental design

This study aims to examine the relationships between rice yields and climatic variables in Taiwan for the 1925-2019 period. Changes in average temperature, diurnal temperature
range (DTR), rainfall, and sunshine duration were considered in the analysis. The analysis also compared warm season and cold season rice yields. The research question is well defined, relevant and meaningful and related to the interests of Peerj.

A literature review was provided in the manuscript to explain previous studies that focused on the relationships between rice yields and climate. Although many studies were cited related to this topic, we notice that no previous studies were cited from Taiwan. I was wondering is any other studies available that examined the rice yield-climate relationships in Taiwan. Also the novelty of this study is not clearly explained. Please clearly indicate that how can this study contribute to the available literature. How different this study would be compared to the studies conducted elsewhere.

I would suggest adding some information to the introduction section regarding the climatic conditions in Taiwan and how they changed from 1925 to date and also about rice yields, if available.

In general methods are described in the manuscript. However, the following should also be considered in the methods section.
--Please explain if management operations (e.g. fertilization, herbicide or pesticide use, types of fertilizers and pesticides, irrigation etc. ) in the farm have every changed since 1925. I would expect that some changes have happened within an about 75 yr period.
--Please explain if any changes occurred in the way meteorological data are collected. For example, please report any changes occurred in the location, types of instruments, etc. I can see from the discussion that the farm changed its location during the period investigated. Please discuss its effect on the collected data.
--Please provide an account of missing data in the climatic and rice yield time series.
--The paper mentions that missing data in climate time series were filled out. Please provide the method used for this purpose and justify it. Again please explain the amount to missing data.
--Please discuss the quality of the data in general. Also discuss the possible source of uncertainty.

Validity of the findings

I would have the following comments regarding the results, dicusssions and conclusions section.
--“Climate change during the rice cropping seasons” section provides some numbers to denote changes between 1975 and 2019. I could not quite understand how these numbers were obtained. For example, in the sentence “The total annual rainfall in the cool and warm cropping seasons decreased by 352 mm and 237 mm”, does these numbers denote that the difference between rainfall measured in 1925 and 2019 or it is the change calculated from the trend of the lines.
-On Figure 1, there are two lines in the average and minimum temperature graphs. Please explain what they mean. Also on Figure 1, some lines are drawn darker than others. Please explain what does color mean?
--Also in this section, I would suggest providing first the general statistics of the data. For example, what is the average temperature during 1927-2019., what is the minimum and maximum during this period, at which year minimum and maximum values occurred. Same should be done for rice yields.
--I would also suggest using a trend analysis technique and reporting the significance of the trends reported for climatic variables and rice yields. I can see that linear trend lines and equations for these lines were reported on Figure 1 . Please extract slopes from these equations and also discuss the statistical significance of the trends. Without the discussion of the statistical significance, the changes reported does not mean anything.
--The limitations of the study should be discussed in the discussion section. Data quality can pose a major limitation on the results.

Reviewer 2 ·

Basic reporting

The authors address the problem of understanding changes in rice production on a Taiwanese farm over a broad time period (1925 to 2019) using climate variables from weather stations that they describe in principle as relevant to rice growth. The analyses are relevant, using time series as well as time intervals. It is important to check the definition of the variables used, in particular, "diurnal temperature range (DTR)" (not sure it is there).
Around line 131. It would be important if the authors can clarify any potential impact of the differences in rice cultivars on the results they obtained.

Experimental design

No comment

Validity of the findings

This study is relevant to better understanding the changes in rice production and could serve as an inspiration for other crops. The study represents an opportunity to understand the relationship between rice phenology and the oscillation of variables throughout the year, specifically in terms of its response to climate change. The underlying data is robust and the conclusions are concrete and overall well-explained.

---

## Round 0.2 · Minor Revisions

I would like to recommend the acceptance of the manuscript, once the authors revised the manuscript according to the comments of the reviewer.

Reviewer 1 ·

Basic reporting

The authors revised the manuscript considering my previous comments and I am satisfied with most of the revisions made.

However, I believe there is still an uncertainity with the units of the trend magnitudes reported in the manuscript. For example, in the sentence "The annual daily average and minimum temperatures calculated from the trend of the regression lines increased by 0.94 and 1.50 °C, respectively, during the cool cropping season and by 1.03 and 1.60 °C, respectively", we cannot understand if the temperatures increased at a rate of 0.94 oC/yr or 1.5 oC/ decade, etc.

Similarly in the sentence "The total annual rainfall calculated from the trend of the regression lines in the cool and warm cropping seasons decreased by 352 and 237 mm, respectively", did precipitation decrease at a rate of 352 mm/yr. (which is very high, so I believe the numbers reported here do not show trend of the regression lines, but something else)

Trend magnitudes should be reported in units of oC/yr or mm/yr. Please refer to the studies in the literature which examined trends in precipitation or air temperatures using linear regresion technique.

Experimental design

No comments

Validity of the findings

No comments

---

## Round 0.3 · accepted · Accept

I recommend the acceptance of this manuscript for publication in PeerJ, thanks.